# URVoice: An Akl-Toussaint/ Graham-Sklansky Approach towards Convex Hull Computation for Sign Language Interpretation

## Abstract

We present URVoice, a vocalizer for the communication impaired, based on the Indian Sign Language Notations. Contemporary psychological theories consider language and speech as devices to understand complex psychological processes and deliver them as cultural products of ideas and communication. Sign and gesture language, offering an intelligent co-ordination of eye-and-hand and ear-and-mouth, has evolved as an intelligent manifestation of speech for the impaired. However, they have very limited modality and iconicity in accommodating a greater range of linguistically relevant meanings. URVoice is an Augmentative and Alternative Communication (AAC) device, which currently features a pipeline of forward communication from signer to collocutor with a novel approach shouldered on convex hull using vision based approach. The solution achieves real time translation of gesture to text/voice using convex hull as the computational geometry, which follows Akl-Toussaint heuristic and Graham-Sklansky scan algorithms. The results are weighed against our other solutions based on conventional Machine Learning and Deep Learning approaches. A futuristic version of URVoice, with voice translated to sign language gestures, will be a complete solution for effectively bridging the cognitive and communication gap between the impaired and the abled lot.

## 1 Introduction

The truism of language and scientific linguistics emerged from man's reinforcing vocal behaviour as a response to the evolving social and natural circumstances. The human species grew to evolve as the singular evolutionary group with a unique neurological organization to support language. Speech and language, that started off as a mapping of meanings to sounds, has now grown to mapping of complex representational intelligence to complicated cognitive communication systems. They have further succeeded in understanding the structural compositions of language in terms of underlying mental expressions.

A communication disorder in human information processing system adversely affects a person's ability to talk, understand, read, and write. Individuals with speech and language impairments lack sufficient representational and communication intelligence and leave them with very less choice to express the forbiddingly abstract levels of subtilies in human communication. Speech and language impairments are considered a high-incidence disability BrainFacts (2012) Francisco (2017) Yukiko & Kiyoshi (2004).

Sign and gesture language, attributed to an intelligent co-ordination of eye-and-hand and ear-and-mouth, evolved as an intelligent manifestation of speech for the impaired. There is no universal sign language used around the world. There are about 138 to 300 different types of sign languages used around the globe todayRichard (2018). A few most widely used sign languages are discussed in detail in Table A1.

However, they have very limited modality and iconicity in accommodating a greater range of linguistically relevant meanings. They also fail to cover the verbal spectrum of temporal and spacial

characteristics of communication. Bridging this gap shall strengthen their mental thoughts, avoid reliance on interpreters, and shall also provide access to new technologies.

## 1.1 TECHNOLOGICAL/MEDICAL SOLUTIONS FOR THE IMPAIRED: STATE OF THE ART

The existing solution includes text-to-speech and sign-to-speech software enabling the speech impaired and the deaf and mute to "speak". These Augmentative and Alternative Communication (AAC) devices (listed in Table A2) range from a simple picture board to a computer program that synthesizes speech from text.

## 1.2 PRESENTING URVOICE, OUR SOLUTION

Considering the limitations of existing AACs, our study focused on designing a cheap, compact and portable vocalizer involving vision-based approaches. We present URVoice, a vocalizer based on the Indian Sign Language Notations. It provides simpler and more intuitive way of communication and makes it possible for remote communication.

URVoice achieves real time translation of gesture to text/voice using convex hull as the computational geometry, which follows Akl-Toussaint heuristic and Graham-Sklansky scan algorithms. The results are weighed against our other solutions based on conventional Machine Learning and Deep Learning approaches. A futuristic version of URVoice, with voice translated to sign language gestures, will be a complete solution for effectively bridging the cognitive and communication gap between the impaired and the abled lot.

## 2 AN ARCHITECTURAL OVERVIEW

### 2.1 URVOICE: ARCHITECTURE

URVoice converts gestures as visual input into audio/ text output for a collocutor or relays as text message to a computer. Similarly, it takes in audio as input from the collocutor/ computer and converts it into gesture/ text as output for the signer. This duplex communication model shall run on an accelerator hardware for optimal performance in real-time communication. A block diagram of the functioning of the vocalizer is presented in the Fig. 1. This paper features a pipeline for one-way communication in URVoice, i.e, visual input to audio/ text output. The gestures are captured in real-time and processed to give audio/ text as output.

### 2.2 CONVEX HULL OPTIMIZATION: A NOVEL APPROACH IN URVOICE

The novelty of URVoice is the use of convex hull as the computational geometry method for recognition of gestures involving use of threshold technique for image segmentation and extraction of various mathematical features from the convex hull. The recognition is accomplished by a simple

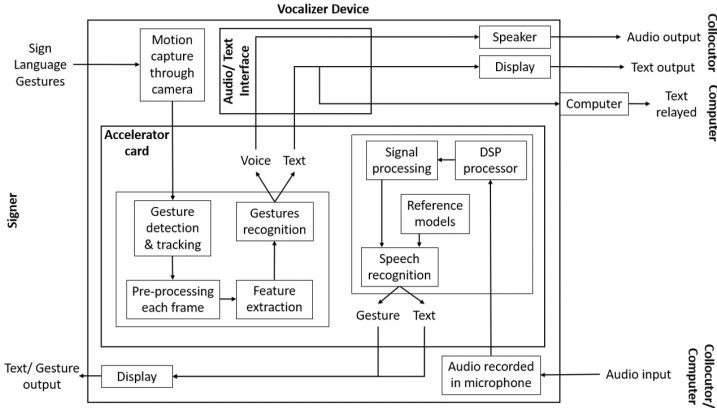

Figure 1: Functional overview of the Vocalizer

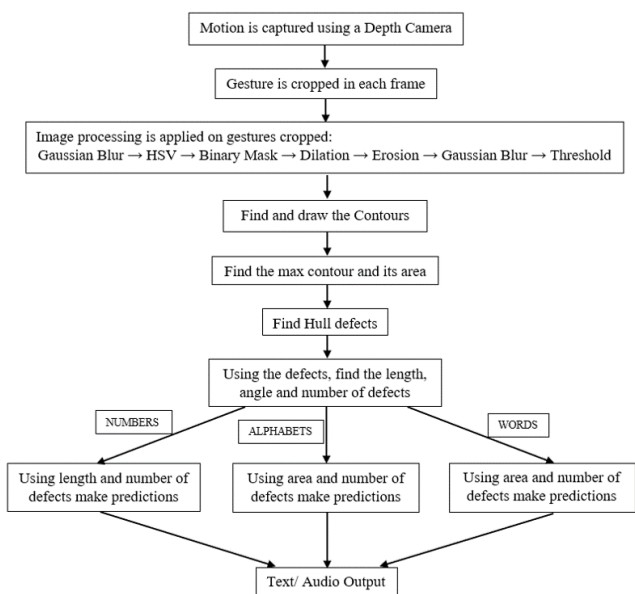

Figure 2: Overview of the algorithm

and efficient rule-based classifier. An overview of the algorithm for hand gesture recognition is described in Fig. 2. This algorithm predicts only static gestures of both single and double hands.

Table 1: Convex Hull Literature

| Algorithm | Proposed by | Complexity | Description |
| --- | --- | --- | --- |
| Graham's method (1972) | Graham | $O(nlogn)$ | To compute the convex hull of n points on the plane based on polar angles |
| Linear time algorithm (1972) | Sklansky Sklansky (1982) | $O(n)$ | It was short and elegant to find the convex hull of a simple polygon in linear time. |
| Gift wrapping method | Chand and Kapur Chand & Kapur (1970), Jarvis Jarvis (1973) | $O(nh)$ | It determines the left most point, which is a vertex of the convex hull, and then searches for the point that lies on one side of the line from the current vertex to it. All the vertices of the convex hull can be found by repeating this procedure. |
| Graham scan | Graham Graham (1972) | $O(nlogn)$ | A modified version of Gift wrapping method. Starting from the lowest point, all the points are sorted in increasing order of the angle of the lowest point made with the x-axis, and then a more efficient searching scheme can be performed. |
| Monotone chain algorithm | Andrew Andrew (1979) | $O(nlogn)$ | A variant of the Gram scan. It sorts points by their coordinates instead of angles. |
| N-dimensional Quickhull | Barber, Dobkin and Huhdanpaa Barber (1996) Hoare (1961) | $O(nlogr)$ | It uses a divide and conquer approach like that of quicksort. |
| Divide and conquer | Preparata and Hong Preparata & Hong (1977) | $O(nlogn)$ | This algorithm is applicable to 3D case. |
| Incremental convex hull algorithm | Kallay Kallay (1984) | $O(nlogn)$ | It is used to develop 3D convex hull algorithms. |
| Ultimate planar convex hull algorithm | Kirkpatrick and Seidel Kirkpatrick & Seidel (1986) | $O(nlogh)$ | The first optimal output-sensitive algorithm. It is a modification of the divide and conquer algorithm by using the technique of marriage-before-conquest and low-dimensional linear programming. |
| Chan's algorithm | Chan Chan (1996) | $O(nlogh)$ | A simpler optimal output-sensitive algorithm. It combines gift wrapping with the execution of an algorithm with $O(nlogn)$ (such as Graham scan) on small subsets of the input. |
| Akl-Toussaint heuristic | Selim Akl and G. T. Toussaint Bhattacharya & Toussaint (1983),Akl & Toussaint (1978) | $O(n)$ | It is often used as the first step in implementations of convex hull algorithm to improve the performance. |
| Graham-Sklansky scan | Preparata & Shamos (1985) | $O(n)$ | It is similar to the Sklansky algorithm. Often used after Akl-Toussaint heuristic |

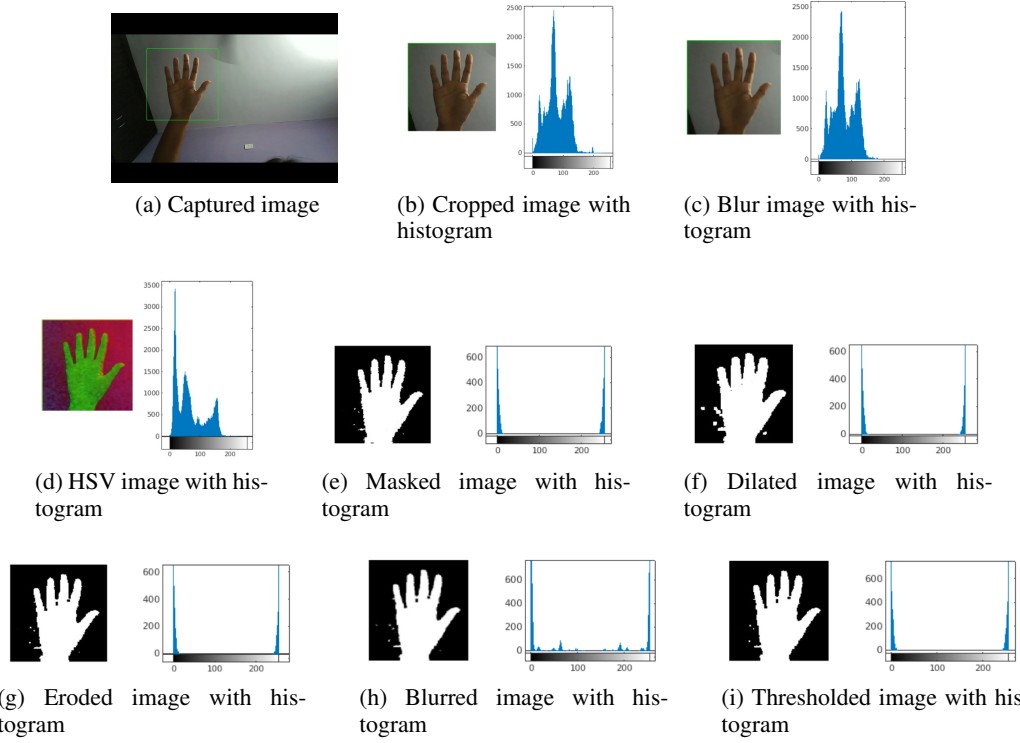

Figure 3: Image Progression in URVoice

# 3 AN ALGORITHMIC OVERVIEW

## 3.1 URVOICE CONVEX HULL OPTIMIZATION

URVoice's central algorithm mainly features convex hull optimization as it provides the necessary keypoints in gestures for their recognition. The computation of the convex hull is a central problem in computational geometry. It has been vastly studied, not only because of its practical applications, such as computer graphics and statistics, but also because other computational geometry problems start with the computation of a convex hull. The various proposed convex hull algorithms are discussed in detail in the Table 1.

### 3.1.1 THE GRAHAM-SKLANSKY/AKL-TOUSSAINT HEURISTIC MODEL

The Graham-Sklansky scan is an important technique in computational geometry which was independently proposed by Graham to compute the convex hull of a sorted set of points and by Sklansky to compute the convex hull of a simple polygon. Whereas the Sklansky scan fails for simple polygons, it succeeds for star-shaped polygons, a fact upon which the correctness of the Graham scan relies. The idea of the Graham scan is to make a single scan through a sorted list of points. At each step in the scan, either a point is deleted or retained based on the test. So, if there are $n$ points, a maximum of $n$ points can be deleted. Thus, the algorithm takes $O(n)$ time. Since its introduction the Graham-Sklansky scan has found widespread application to other problems. The idea of Akl-Toussaint heuristic is to quickly exclude many points that would not be part of the convex hull.Xianshu et al. (2003)

Thus, the convex hull method follows Akl-Toussaint heuristic, which results in a monotone polygon, followed by Graham-Sklansky scan to obtain the points of convex hull.

## 3.2 URVOICE COMPUTATIONAL PIPELINE

The pipeline presented consists of the following stages :-

1. Image Acquisition
2. Pre-processing
3. Feature Extraction
4. Gesture Prediction

It is discussed in detail in the following subsections. The mathematical model for the pipeline is described in A.1.

### 3.3 IMAGE ACQUISITION

An RGB image of gesture is captured during run-time with the help of an external depth camera, as shown in Fig. 3(a). The camera used in this research is Orbbec Astra Pro with a resolution of 1280*720 @30fps.

### 3.4 PRE-PROCESSING

The region of interest is cropped out of the captured image having a size of $(200 \times 200)$ pixels for number gestures and $(400 \times 200)$ pixels for alphabet and word gestures, and is stored as $crop\_image$, as shown in Fig. 3(b) along with its histogram. The image is pre-processed using the

Table 2: Pre-processing stages

| Stage | Name of filter | Description |
|---|---|---|
| 1 | Gaussian Blur | Gaussian blur is applied first on $crop\_image$ to smoothen the pixel values of the hand so as to create some amount of uniformity. Here, a Gaussian filter convolves over this image and thereby blurs it. The workflow for obtaining the Gaussian filter is presented in Algorithm 1 in A.2. From this algorithm, the value of $\sigma$ is obtained as 0.8 and the Gaussian filter obtained is: $$\begin{bmatrix} 0.06292 & 0.124998 & 0.06292 \\ 0.124998 & 0.248326 & 0.124998 \\ 0.06292 & 0.124998 & 0.06292 \end{bmatrix}$$ After convolution, a blur image is obtained, as shown in Fig. 3(c), and is stored as $blur$ with BGR pixel format. The histogram of Fig. 3(c) shows minor changes in the peaks and valleys of the histogram. The BGR format of the image is then converted to HSV format to mainly focus on the intensity component of the image which helps in eliminating the background noise and shadows. |
| 2 | HSV | HSV color space is most suitable for color-based image segmentation and it is and it is hence directly applied on $blur$ with BGR image format. The Algorithm 2 A.2 presents the workflow for obtaining the hsv values. The bgr values of the blurred image are then replaced by the hsv values after applying Algorithm 2 A.2 and is stored in $hsv$. The resultant image obtained is as shown in Fig. 3(d) along with its histogram. It can be observed from the image that the region of interest, i.e., the hand, is now covered with green pixels. Thus, a mask has to be applied on hsv to focus on the region of interest. |
| 3 | Mask | A mask is applied on $hsv$ to crop out the gesture within the image. The workflow for the same is presented in Algorithm 3 A.2. A binary mask is returned and stored in $mask$. The resultant image obtained is as shown in Fig. 3(e) along with its histogram which now has two peaks. |
| 4 | Dilation | Dilation is applied on $mask$ to overcome the disfigured regions and also assist in joining broken parts of the image. It is observed that the thickness of the foreground object has increased. This operation consists of convolving the image ($mask$) with a kernel ($5 \times 5$ unit matrix). The kernel has a defined anchor point which is at the center. As the kernel is scanned over the image, the maximal pixel value overlapped by the kernel is computed and this replaces the image pixel at the anchor point position. This process takes place once throughout the image. The final image obtained, as shown in Fig. 3(f) along with its histogram, is stored in $dilation$. |
| 5 | Erosion | Erosion is applied on $dilation$ to eliminate noise if any. It is observed that the thickness of the foreground object has decreased. This operation is similar to dilation. As the kernel [5x5 unit matrix] is scanned over the image, it computes the minimal pixel value overlapped by the kernel and this minimal value replaces the image pixel under the anchor point. This process takes place once throughout the image. The final image obtained, as shown in Fig. 3(g) along with its histogram, is stored in $erosion$. Now, the resultant image contains rough edges which has to be fixed by blurring it. |
| 6 | Gaussian Blur | Gaussian blur is again applied on $erosion$ to smoothen the image. This helps to remove the deformities on the edges of the hand gesture formed due to erosion. The procedure followed is same as in Point 1 under Pre-processing and the resultant image obtained, as shown in Fig. 3(h), is stored in $filtered$. The histogram in the Fig. 3(h) shows appearance of small new peaks between the earlier obtained peaks. This is considered as noise and hence has to be removed by thresholding the image. |
| 7 | Threshold | Thresholding is applied on $filtered$ to ensure uniformity of higher-grade gray pixel values in the image. The workflow for thresholding the image is presented in Algorithm 4 A.2. The resultant binary image obtained, as shown in Fig. 3(i) is stored in $thresh$. The small peaks in the histogram of the blur image (Fig. 3(h)) are erased in the histogram of the thresholded image (Fig. 3(i)). |

Table 3: Gesture Features

| Feature | Description |
|---|---|
| Contours | Contours algorithm processes the arbitrary binary image $thresh$ and returns a vector of detected contour points. It uses CV_RETR_TREE mode to retrieve all the contours and reconstruct a full hierarchy of nested contours along with CV_CHAIN_APPROX_SIMPLE method to compress horizontal, vertical, and diagonal segments and leaves only their end points. Contour retrieval follows Suzuki's algorithm which is explained as follows: The function $f(x, y)$ denotes the value of the pixel at location $(x, y)$. The uppermost row, the lowermost row, the rightmost column, and the leftmost column of the picture composes the frame. Then, a unique number is assigned to every new border and it is denoted by NBD. The criteria for checking the outer border or hole border is shown in Fig. 4(a). The NBD of the frame is assumed to be 1. Other borders are numbered sequentially. The information of the parent of any border is saved in LNBD or last NBD. The workflow to find the contour points is presented in Algorithm 5 A.2. The retrieved tree of list of points are stored in $contours$. Using $contours$, the largest contour in terms of area is isolated and stored as $contour$. Fig. 4(b) shows the contour line (as green line) joining all the points in $contour$. The area of $contour$ is calculated using the formula presented in Algorithm 6 A.2. |
| Convex Hull and Convexity Defects | The smallest convex polygon, that encloses all the set of contour points in $contour$, is found by convex hull method which follows Akl-Toussaint heuristic followed by Graham-Sklansky scan as presented in Algorithm 1. This algorithm returns the sequence of stack of vertices of the convex hull and is stored in $hull$. Fig. 4(b) shows the convex hull lines (as red lines) joining all the points in $hull$. Considering $H[n]$ as convex hull points and $C[n]$ as contour points, for each pair of adjacent hull points ($H[i]$, $H[i+1]$), defining one edge of the convex hull, the distance from the edge for each point on the contour $C[n]$ that lies between $H[i]$ and $H[i+1]$ (excluding $C[n] == H[i+1]$) is calculated. The workflow for calculating the maximum distance is presented in Algorithm 7 A.2. If the distance is greater than zero, then a defect is considered to be present. When a defect is present, the value of $i$, $i+1$, the index ($n$) of the contour point where the maximum is located and the maximum distance is recorded in $defects$ as a list containing: [ start point, end point, farthest point, approximate distance to farthest point ]. |
| Length and Angle | For each defect in $defects$, the coordinates of start point, end point and farthest point is stored in $start$, $end$ and $far$ respectively. The workflow for calculating lengths of the sides of the triangle formed by joining these points and the angle $\theta$ formed at the vertex of the farthest point is presented in Algorithm 8 A.2. A depiction of these distances and angle with respect to fingers having convex defect is shown in Fig. 4(c). If the $angle$ calculated is less than $90°$, then a variable $count\_defects$ is incremented by one. The final image depicting the contours in green line and defects in red dots is shown in Fig. 4(d). |

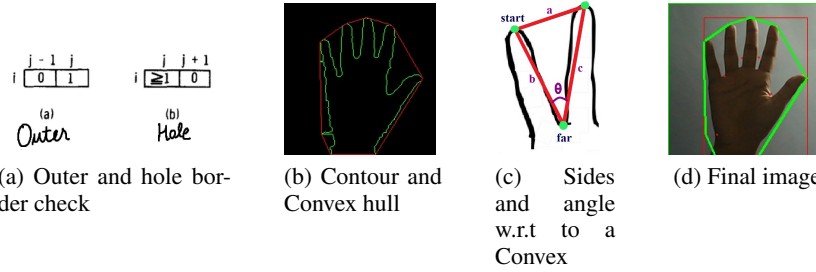

(a) Outer and hole border check

(b) Contour and Convex hull

(c) Sides and angle w.r.t to a Convex defect

(d) Final image

Figure 4: Convex Hull Optimization in URVoice

Table 4: Gesture prediction

| Gesture | Methodology |
|---|---|
| Number | When a defect with its vertex angle less than $90°$ is encountered, a variable $A$ is incremented with the value of the distance $a$, else, a variable $B$ is incremented with the value of the distance $b$. These variables $A$ and $B$ represents the sum of distances between the fingers and the sum of lengths of the fingers. Finally, the predictions for the number gestures are obtained by using the values of $count\_defects$, $A$ and $B$ as presented in Algorithm 9 A.2. |
| Alphabet | The alphabet gestures are predicted with the help of $count\_defects$ and $area$ values as presented in Algorithm 10 A.2. |
| Word | The word gestures are predicted with the help of $count\_defects$ and $area$ values as presented in Algorithm 11 A.2. |

OpenCV library in Python to extract the necessary features. The pre-processing stages are explained in Table 2.

## 3.5 FEATURE EXTRACTION

By employing the fully segmented image $thresh$, required features are extracted and assembled for prediction. The requisite features for prediction are presented in Table 3.

---

**Algorithm 1** : Convex Hull(*contour*)

1: **// Purpose:** To find the coordinates of convex hull of the *contour*
2: **// Input:** $(x, y)$ coordinates of *contour*
3: **// Output:** $(x, y)$ coordinates of boundary points of convex hull
4: **// Parameters:** $TopMost$- point at the top of $HullStack$, $TopLess$- point just below $TopMost$
5: -----------------------------------------------------------------
6: $p_{minX}$ = point with lowest X-coordinate
7: $p_{maxX}$ = point with highest X-coordinate
8: $p_{minY}$ = point with lowest Y-coordinate
9: $p_{maxY}$ = point with highest Y-coordinate
10: Define the bounding box and quadrant of $p_{minX}, p_{maxX}, p_{minY}$ and $p_{maxY}$
11: **if** the points lie inside the quadrant **then**
12:     Discard the points
13: **else**
14:     Select the point lying in the regions between the bounding box and quadrant
15: **end if**
16: **for** each corner region **do**
17:     $p_1 = min\{f(X, Y) = X - Y\}$
18:     Discard points lying inside the triangle $(p_{minX}, p_1, p_{pmaxY})$
19: **end for**
20: **if** the non-discarded points lie on and above the line $L(p_{minX}, p_{maxX})$ **then**
21:     Sort them in ascending order of their X-coordinates and stack in P
22: **else**
23:     Sort them in descending order of their X-coordinates and stack in P
24: **end if**
25: Apply Graham-Sklansky procedure to the obtained monotone polygon:
26: Initialize empty HullStack
27: $P[N + 1] := P[1]$
28: Push $(P[1], HullStack)$
29: Push $(P[2], HullStack)$
30: **for** $Ind := 3$ to $N + 1$ **do**
31:     **while** ($HullStack$ stacks more than one element) and ($P[Ind]$ is to the left of $L(TopLess, TopMost)$ **do**
32:         pop($HullStack$) [discard $TopMost$]
33:     **end while**
34:     push($P[Ind], HullStack$)
35: **end for**
36: pop($HullStack$)

---

Table 5: Gestures studied

| Gesture | Names |
|---|---|
| Numbers | 1, 2, 3, 4, 5, 6, 7, 8, 9 |
| Alphabets | B, C, L, P, R, W, T |
| Words | Doctor, This, Your, Mine |

Table 6: Time taken to process the code

| Gesture | Mid-range of time taken (in seconds) | |
|---|---|---|
| | Text output | Audio output |
| Numbers | 0.1221 | 1.1133 |
| Alphabets | 0.1103 | 1.1362 |
| Words | 0.0804 | 1.1457 |

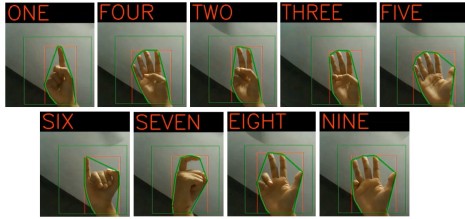

(a) Number predictions displayed on screen

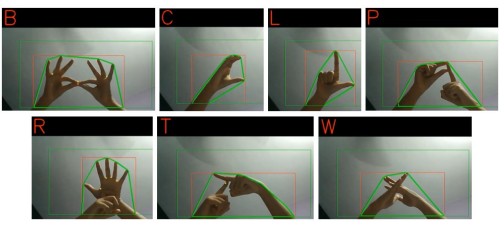

(b) Alphabet predictions displayed on screen

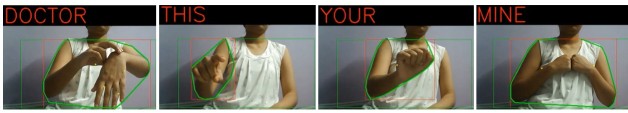

(c) Word predictions displayed on screen

Figure 5: Output

Table 7: Benchmarking Convex Hull Optimization with selected ML/DL Models

| Model | Classes trained | Dataset Specifications | Model Architecture | Performance Analysis |
|---|---|---|---|---|
| Support Vector Classifier | 0, 1, 2, 3, 4, 5, 7, 8, 9 [Total:9 classes] | RGB image dataset of static single hand gestures of shape [400x400x3]. 300 images per class [Total: 2700 images]. Training set: 2518 images Validation set: 169 images per class Testing set: 13 images per class | A pipeline with Randomized-PCA for dimensionality reduction followed by SVC classifier with "rbf" kernel and "balanced" class_weight. Trained data using GridSearchCV. | Testing accuracy: 95.86% Testing loss: 4.14% Validation accuracy: 93.68% Validation loss: 6.32% |
| Artificial Neural Network (only numbers) | 1, 2, 3, 4, 5, 6, 7, 8, 9 [Total: 9 classes] | Numerical dataset of 8 extracted features of hand gestures [normalised data]. 1000x8 features per class [Total: 9000x8 features]. Training set: 6300x8 features per class Testing set: 1800x8 features per class | A Sequential Model with 3 hidden layers which uses relu activation function. The output layer uses softmax activation function, adam optimiser and categorical crossentropy as loss function. | Training accuracy: 94.59% Training loss: 17.97% Validation accuracy: 97.33% Validation loss: 10.94% |
| Artificial Neural Network (numbers, alphabets and words) | 1, 2, 3, 4, 5, 6, 7, 8, 9, B, C, L, P, R, W, T, Doctor, This, Your, Mine [Total: 20 classes] | Numerical dataset of extracted features of hand gestures [normalised data]. 1000x8 features per class [Total: 20000x8 features]. Training set: 6300x8 features per class Testing set: 1800x8 features per class | A Sequential Model with 4 hidden layers which uses relu activation function. The output layer uses softmax activation function, adam optimiser and categorical crossentropy as loss function. | Training accuracy: 93.70% Training loss: 17.80% Validation accuracy: 95.32% Validation loss: 14.01% |
| Convolutional Neural Network (1) | 0, 1, 2, 3, 4, 5, 7, 8, 9 [Total:9 classes] | RGB image dataset of static single hand gestures of shape [400x400x3]. 60 images per class [Total: 540 images]. Training set: 40 images per class Testing set: 20 images per class | A Sequential Model with 2 Convolution2D layers, 2 Max-Pooling2D layers and 2 hidden layers of neural network with relu as activation function. The output layer uses softmax activation function, adam optimiser and categorical crossentropy as loss function. | Accuracy: 86.67% Loss: 13.33% Precision: 91% Recall: 87% F1-score: 87% |
| Convolutional Neural Network (2) | 0, 1, 2, 3, 4, 5, 7, 8, 9 [Total:9 classes] | RGB image dataset of static single hand gestures of shape [400x400x3]. 300 images per class [Total: 2700 images]. Training set: 250 images per class Validation set: 30 images per class Testing set: 20 images per class | A Sequential Model with 2 Convolution2D layers, 2 Max-Pooling2D layers and 2 hidden layers of neural network with relu as activation function. The output layer uses softmax activation function, adam optimiser and categorical crossentropy as loss function. | Accuracy: 98.89% Loss: 2.46% Precision: 98% Recall: 98% F1-score: 98% |
| Convex Hull Optimization technique | 1, 2, 3, 4, 5, 6, 7, 8, 9, B, C, L, P, R, W, T, Doctor, This, Your, Mine [Total:20 classes] | Real-time RGB images in video stream: [200x200] pixels for number gestures and [400x200] pixels for alphabet and word gestures. | A rule based classifier as discussed in subsection 3.6 | Refer Table 6 |

## 3.6 GESTURE PREDICTION

The gestures of the Indian sign language studied are mentioned in the Table 5. By employing the extracted features along with additional values, predictions are made for gestures of numbers, alphabets and words as described in Table 4.

## 4 RESULTS AND DISCUSSION

### 4.1 URVOICE RESULTS

The algorithm relays the prediction both in terms of text and audio. The algorithm with output as text uses $cv2$ library of OpenCV to display the predictions on the screen. Fig. 5(a), 5(b), 5(c) shows the text output of various gestures displayed on the screen. Likewise, the algorithm with audio as

output makes use of $pyttsx3$ library. Once the predictions are made, the text is instantaneously converted to speech. One advantage of using Python's $pyttsx3$ library is that it works offline and hence it is convenient to use anywhere. The mid-range of the time taken to process the code to obtain the output of prediction is provided in Table 6. The data in the table illustrates the fact that the time required for audio output is greater than text output due to the text to speech conversion module and also there is a small lag in the run-time due to the time spent to play the audio.

## 4.2 BENCHMARKING CONVEX HULL OPTIMIZATION WITH SELECTED ML/DL MODELS

We trained and deployed a few other Machine learning and Deep learning models for hand gesture recognition, the details of which are listed in Table 7. For this, we created two different datasets:

1. Image Dataset: 2700 RGB images of 3 individuals were captured for 9 classes of number gestures with different brightness levels.
2. Numerical Dataset: Extracted 8 real-time hand gesture features of 2 individuals which includes: sum of distances between fingers, sum of lengths of fingers from tip to convex defects on both sides, sum of distances from tip of fingers to the centroid of the palm, sum of angles between the fingers, total area of hand gesture covered, total perimeter covered by the convex points of the hand gesture and number of convex defects. These features were extracted using the same algorithm discussed in this paper.

Although these models performed well, the convex hull optimization technique was preferred over this based on the obtained performance and the type of system we wanted to implement. Using a CNN model would result in designing a GPU based system for hand-held device which is not a feasible solution due to factors like memory usage, size and cost. The experiments performed using convex hull method deduces best results with the features under study and also was able to reliably recognize the gestures in real-time, though there were some limitations imposed by the presence of image noise. Thus, the convex hull optimization technique proves to have better static gesture recognition rates.

## 5 CONCLUSION AND FUTURE WORK

This paper presents the architectural design of the vocalizer, URVoice for Indian sign language interpretation. This device comes under the category of Augmentative and Alternative Communication (AAC) Devices. We have described and evaluated the forward communication process of converting static gesture input to text/ audio output. The novelty of this approach is the use of convex hull as the computational geometry method for recognition of gestures. We have also trained a CNN model for predicting gestures from images and ANN model for predicting gestures from numerical data. We preferred convex hull over CNN due to its computational efficiency and architecture.

In our future work, we will build an algorithm to recognize the dynamic gestures and implement the feedback channel of converting audio input from a collocutor/ computer to text/ gesture output which will be displayed on the screen to complete the software part of our architecture. We will also be developing a hardware prototype using accelerator hardware and embedded firmware. This will serve to be a high-performance, low cost and portable infrastructure to be used as a gadget that serves as the interpreter and translator for the communication impaired.

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

## A   APPENDIX

### A.1   MATHEMATICAL MODEL

A discrete RGB image is defined as m × n × k hypermatrix $I$, where the $mnk^{th}$ entry represents the $mn^{th}$ pixel in colour channel $k$. Consider $(k = 1)^{th}$ channel as $B$ matrix, $(k = 2)^{th}$ channel as $G$ matrix and $(k = 3)^{th}$ channel as $R$ matrix. A Gaussian function $g_t(x, y)$ is applied separately to each $k^{th}$ matrix where,

$$g_t(x, y) = \frac{1}{2\pi\sigma^2} * e^{-\frac{x^2+y^2}{2\sigma^2}} // \sigma - Standard\ deviation \tag{1}$$

Then $b_t$, a Gaussian blur of image $I_k$, is achieved by convolving it with $g_t(x, y)$,

$$b_t(x, y) = I_k(x, y) * g_t(x, y) = \int_{-\infty}^{\infty}\int_{-\infty}^{\infty} g_t(x - \alpha, y - \beta)I_k(\alpha, \beta)d\alpha d\beta \tag{2}$$

The blur image $B_{(i,j,k)}$ from $I$ matrix is obtained by:

$$B_k(i, j) = \frac{\sum_{p=1}^{m}\sum_{l=1}^{n} g_t(p, l) \cdot I_k(i + p - 1, j + l - 1)}{\sum_{p=1}^{m}\sum_{l=1}^{n} g_t(p, l)} \tag{3}$$

$B_k(i, j)$ values are normalized and transformed into HSV matrix with hue $H \in [0°, 360°]$, saturation $S \in [0, 1]$ and value $V \in [0, 1]$. The equations for conversion of $BGR$ matrix into $HSV$ matrix is presented in algorithm 2 A.2. A new binary matrix with values 0 or 255 is created from $HSV$ as defined below:

$$M(i, j) = \begin{cases} 255, & \begin{aligned} &\text{for } (2 \leq H(i, j) \leq 20) \wedge \\ &(0 \leq S(i, j) \leq 255) \wedge \\ &(0 \leq V(i, j) \leq 255) \end{aligned} \\ 0, & \text{otherwise} \end{cases} \tag{4}$$

The intensity of white pixel is intensified by dilation using a $5 \times 5$ unit matrix $K$. Dilation of matrix $M$ by $K$ is defined by:

$$D = M \oplus K = \bigcup_{k \in K} M_k = \{x : \hat{K}_x \cap M \neq 0\} \tag{5}$$

where,

$$\hat{K} = \{x : x = -k, for\ k \in K\} \tag{6a}$$
$$K_x = \{c : c = a + x, for\ a \in K\} \tag{6b}$$

In matrix transformation, dilation is implemented as:

$$D(i, j) = \{x : max[K(p, l) \cdot M(i + p - 1, j + l - 1)]\} \tag{7}$$

where, $p \in (1, m)\ and\ l \in (1, n)$

The intensity of white pixel is reduced by erosion using a $5 \times 5$ unit matrix $K$. Erosion of matrix $D$ by $K$ is defined by:

$$E = D \ominus K = \bigcap_{k \in K} D_{-k} = \{x : K_x \subseteq D\} \tag{8}$$

In matrix transformation, erosion is implemented as:

$$E(i, j) = \{x : min[K(p, l) \cdot D(i + p - 1, j + l - 1)]\} \tag{9}$$

where, $p \in (1, m)\ and\ l \in (1, n)$

Again, Gaussian function is applied to matrix $E$ and is defined as:

$$f(x, y) = \int_{-\infty}^{\infty} \int_{-\infty}^{\infty} g_t(x - \alpha, y - \beta) E(\alpha, \beta) d\alpha d\beta \tag{10a}$$

$$F(i, j) = \frac{\sum_{p=1}^{m} \sum_{l=1}^{n} g_t(p, l) \cdot E(i + p - 1, j + l - 1)}{\sum_{p=1}^{m} \sum_{l=1}^{n} g_t(p, l)} \tag{10b}$$

A threshold value is set to classify the values of matrix $F$ to create a binary matrix of value 0 or 1. The threshold value is defined by:

$$T = \frac{\mu_b + \mu_o}{2} \tag{11}$$

where, $\mu_b$ and $\mu_o$ are the means of the distribution in the gray level values in matrix $F$. The matrix obtained after thresholding $F$ is defined by,

$$T(i, j) = \begin{cases} 0, & \text{if } F(i, j) \leq T \\ 1, & \text{otherwise} \end{cases} \tag{12}$$

A curve joining points with equal intensities in the matrix $T$ would represent contours. The (i, j) coordinates of contours are obtained as presented in algorithm 5 A.2.

Let $C(x, y)$ be the contour line joining all contour points, then the area covered by the line is given by the Green's theorem as:

$$Area = \frac{1}{2} \int_C xdy - ydx \tag{13}$$

In coordinate system,

$$Area = \sum_{k=0}^{n} \frac{(x_{k+1} + x_k)(y_{k+1} - y_k)}{2} \tag{14}$$

The convex hull of a set of points C in n dimensions is the intersection of all convex sets containing C. For N points $p_1, \ldots p_N$, the convex hull $H$ is given by the expression:

$$H = \{\sum_{j=1}^{N} \lambda_j p_j : \lambda_j \geq 0\ for\ all\ j\ and\ \sum_{j=1}^{N} \lambda_j = 1\} \tag{15}$$

Let the line joining the hull points be $\vec{H}$ and the perpendicular line from $\vec{H}$ to the contour line lying within it be $\vec{d}$. If $\vec{H} \times \vec{d}$ is maximum, then a defect is considered and the coordinates of the contour point, hull points and the distance are recorded. The distances, angle and defects using these points are calculated as in Algorithm 8 A.2 to make various predictions.

## A.2   ALGORITHMS

---

**Algorithm 1** : GaussianBlur(crop_image, (3,3), 0)

---

1: **// Purpose:** To obtain the Gaussian filter
2: **// Input:** crop_image, kernel_size, sigmaX value
3: **// Output:** Gaussian filter
4: **// Parameters:** $x$ = distance from the origin in the horizontal axis
5: $y$ = distance from the origin in the vertical axis
6: $\sigma$ = the standard deviation of the Gaussian distribution
7: - - - - - - - - - - - - - - - - - - - - - - - - - - - - - - - - - - - - - - - - - - - - - - - - - - - - - - - - -
8: **if** $sigmaX = 0$ **then**
9:      $\sigma = 0.3 * ((ksize - 1) * 0.5 - 1) + 0.8$          // ksize = kernel_size(3)
10: **end if**
11: $G(x,y) = \frac{1}{2\pi\sigma^2} * e^{-\frac{x^2+y^2}{2\sigma^2}}$

---

---

**Algorithm 2** : HSV($blur$)

---

1: **// Purpose:** To obtain the HSV image
2: **// Input:** $blur$ with BGR values
3: **// Output:** HSV image
4: **// Parameters:** b,g,r = Coordinate pixel values of BGR image
5: - - - - - - - - - - - - - - - - - - - - - - - - - - - - - - - - - - - - - - - - - - - - - - - - - - - - - - - - -
6: **for** each coordinate pixel value (b,g,r) **do**
7:      Divide b, g, r by 255
8:      $cmax = max(b, g, r)$
9:      $cmin = min(b, g, r)$
10:      $difference = cmax - cmin$
11:      **if** cmax = cmin = 0 **then**
12:          $h = 0$
13:      **end if**
14:      **if** cmax = r **then**
15:          $h = (60 * ((g - b)/difference) + 360)\%360$
16:      **end if**
17:      **if** cmax = g **then**
18:          $h = (60 * ((b - r)/difference) + 120)\%360$
19:      **end if**
20:      **if** cmax = b **then**
21:          $h = (60 * ((r - g)/difference) + 240)\%360$
22:      **end if**
23:      **if** cmax = 0 **then**
24:          $s = 0$
25:      **end if**
26:      **if** cmax != 0 **then**
27:          $s = (diff/cmax) * 100$
28:      **end if**
29:      $v = cmax * 100$
30: **end for**

---

---

**Algorithm 3** : Mask($hsv$, lower limit range, upper limit range)

---

1: **// Purpose:** To mask $hsv$
2: **// Input:** $hsv$, lower limit range(2, 0, 0), upper limit range(20, 255, 255)
3: **// Output:** Binary $mask$
4: **// Parameters:** lowerb = lower pixel limit range
5: upperb = upper pixel limit range
6: hsv(I) = hsv value
7: mask = destination image
8: - - - - - - - - - - - - - - - - - - - - - - - - - - - - - - - - - - - - - - - - - - - - - - - - - - - - - - - - - - - - -
9: **for** each coordinate hsv pixel value **do**
10:     **if** $(lowerb_0 \leq hsv(I)_0 \leq upperb_0) \wedge (lowerb_1 \leq hsv(I)_1 \leq upperb_1) \wedge (lowerb_2 \leq hsv(I)_2 \leq upperb_2)$ **then**
11:         $mask = 255$ (white)
12:     **else**
13:         $mask = 0$ (black)
14:     **end if**
15: **end for**

---

**Algorithm 4** : Threshold(filtered)

---

1: **// Purpose:** To obtain threshold of $filtered$
2: **// Input:** $filtered$
3: **// Output:** $thresh$
4: **// Parameters:** f (x, y) = Coordinate Pixel Value of $filtered$
5: T = Threshold Value(127)
6: g(x,y) = destination pixel value of $thresh$
7: - - - - - - - - - - - - - - - - - - - - - - - - - - - - - - - - - - - - - - - - - - - - - - - - - - - - - - - - - - - - -
8: **for** coordinate pixel value **do**
9:     **if** $f(x, y) \leq T$ **then**
10:         $g(x, y) = 0$
11:     **else**
12:         $g(x, y) = 1$
13:     **end if**
14: **end for**

---

---

**Algorithm 5** : Contours

---

1: **// Purpose:** To find contour points
2: **// Input:** $thresh$
3: **// Output:** $contours$
4: - - - - - - - - - - - - - - - - - - - - - - - - - - - - - - - - - - - - - - - - - - - - - - - - - - - - - - - - - - -
5: Scan the image from left to right till an object pixel is found
6: Check if the pixel is an outer border or hole border.
7: When a new row is scanned, reset $LNBD$ to 1
8: **if** $pixels > 0$ **then**
9:     **if** $f_{xy} = 1$ and $f_{x,y-1} = 0$ [outer border] **then**
10:         $NBD+ = 1$
11:         Set $(x_2, y_2)$ as $(x, y - 1)$
12:     **else if** [hole border] **then**
13:         $NBD+ = 1$
14:         Set $(x_2, y_2)$ as $(x, y + 1)$
15:         **if** $f_{xy} > 1$ **then**
16:             $LNBD = f_{xy}$
17:         **end if**
18:     **else**
19:         Go to line 42
20:     **end if**
21:     From this starting point, trace the border
22:     Starting from $(x_2, y_2)$ check clockwise around the pixels in the neighbourhood of $(x, y)$. When a nonzero pixel is found, denote it as $(x_1, y_1)$
23:     **if** no nonzero pixels are found **then**
24:         Set $f_{xy} = -NBD$
25:         Go to line 29
26:     **end if**
27:     Set $(x_2, y_2) = (x_1, y_1)$ and $(x_3, y_3) = (x, y)$
28:     Starting from the next element of the pixel $(x_2, y_2)$ in the counter-clockwise order, traverse the neighbourhood of the $(x_3, y_3)$ in the counter-clockwise direction to find the first nonzero pixel and set it to $(x_4, y_4)$.
29:     **if** pixel at $(x_3, y_3 + 1)$ is a 0-pixel belonging to the region outside the boundary **then**
30:         The current pixel $(x_3, y_3) = -NBD$
31:     **else if** the pixel at $(x_3, y_3 + 1)$ is not a 0-pixel and the current pixel value is 1 **then**
32:         The current pixel $(x_3, y_3) = NBD$
33:     **else**
34:         Do not change the current pixel value
35:     **end if**
36:     **if** In line 28, $(x_4, y_4) = (x, y)$ and $(x_3, y_3) = (x_1, y_1)$ [back to starting point] **then**
37:         Go to line 42
38:     **else**
39:         Set $(x_2, y_2) = (x_3, y_3)$ and $(x_3, y_3) = (x_4, y_4)$
40:         Go back to line 28
41:     **end if**
42:     **if** $f_{xy} != 1$ **then**
43:         Set $LNBD = |f_{xy}|$
44:     **end if**
45:     Start scanning from the next pixel $(x, y + 1)$
46:     Stop the process when the bottom right corner of the image is reached.
47: **end if**

---

---

**Algorithm 6** : Area(contour)

---

1: **// Purpose:** To find the area enclosed by the contour
2: **// Input:** $contours$
3: **// Output:** $Area$
4: **// Parameters:** n = number of vertices
5: $(x_k, y_k)$ = kth point when labelled in a counter-clockwise manner (the co-ordinates are stored in the contour list)
6: $(x_{n+1}, y_{n+1}) = (x_0, y_0)$: the starting vertex is found both at the start and end of the list of vertices.
7:
8: - - - - - - - - - - - - - - - - - - - - - - - - - - - - - - - - - - - - - - - - - - - - - - - - - - - - -
9: $Area = \sum_{k=0}^{n} \frac{(x_{(k+1)} + x_k)(y_{(k+1)} - y_k)}{2}$

---

**Algorithm 7** : Max Distance($hull$, $contour$)

---

1: **// Purpose:** To find the maximum distance between convex hull edge and contour
2: **// Input:** $hull$, $contour$
3: **// Output:** Maximum distance
4: **// Parameters:** $H[n]$ = convex hull points
5: $C[n]$ = contour points
6: - - - - - - - - - - - - - - - - - - - - - - - - - - - - - - - - - - - - - - - - - - - - - - - - - - - - -
7: **for** i = 0 to n **do**
8: $\quad dx_0 = H[i+1]_x - H[i]_x$
9: $\quad dy_0 = H[i+1]_y - H[i]_y$
10: $\quad$ **if** $(dx_0 = 0) \wedge (dy_0 = 0)$ **then**
11: $\quad\quad scale = 0$
12: $\quad$ **else**
13: $\quad\quad scale = \frac{1}{\sqrt{dx_0 * dx_0 + dy_0 * dy_0}}$
14: $\quad$ **end if**
15: $\quad$ **for** j=0 to n **do**
16: $\quad\quad dx = C[j]_x - H[i]_x$
17: $\quad\quad dy = C[j]_y - H[i]_y$
18: $\quad\quad distance = \mid (-dy_0 * dx + dx_0 * dy) \mid *scale$
19: $\quad$ **end for**
20: $\quad distance = max(distance)$
21: **end for**

---

**Algorithm 8** : Calculate lengths and angles

---

1: **// Purpose:** To calculate the lengths and angles
2: **// Input:** $start$, $end$ and $far$
3: **// Output:** $a$, $b$, $c$ and $angle$
4: - - - - - - - - - - - - - - - - - - - - - - - - - - - - - - - - - - - - - - - - - - - - - - - - - - - - -
5: $a = \sqrt{(end[0] - start[0])^2 + (end[1] - start[1])^2}$
6: $b = \sqrt{(far[0] - start[0])^2 + (far[1] - start[1])^2}$
7: $c = \sqrt{(end[0] - far[0])^2 + (end[1] - far[1])^2}$
8: $angle = sin^{-1}\left(\frac{(b^2 + c^2 - a^2) * 180}{2bc * 3.14}\right)$

---

---

**Algorithm 9** : Number prediction

---

1: **// Purpose:** To predict the numbers
2: **// Input:** $angle$, $A$- sum of lengths of $a$ and $B$- sum of lengths of $b$
3: **// Output:** Predicted number
4: - - - - - - - - - - - - - - - - - - - - - - - - - - - - - - - - - - - - - - - - - - - - - - - - - - - - - - - -
5: **if** $count\_defects = 0 \ \wedge \ B \leq 190$ **then**
6:    $pred$ = "ONE"
7: **else if** $count\_defects = 0 \ \wedge \ B \geq 190$ **then**
8:    $pred$ = "SIX"
9: **else if** $count\_defects = 1 \ \wedge \ A \leq 40$ **then**
10:    $pred$ = "TWO"
11: **else if** $count\_defects = 1 \ \wedge \ A \geq 40$ **then**
12:    $pred$ = "SEVEN"
13: **else if** $count\_defects = 2 \ \wedge \ A \leq 70$ **then**
14:    $pred$ = "THREE"
15: **else if** $count\_defects = 2 \ \wedge \ A \leq 80$ **then**
16:    $pred$ = "EIGHT"
17: **else if** $count\_defects = 3 \ \wedge \ A \leq 90$ **then**
18:    $pred$ = "FOUR"
19: **else if** $count\_defects = 3 \ \wedge \ A \leq 100$ **then**
20:    $pred$ = "NINE"
21: **else if** $count\_defects = 4$ **then**
22:    $pred$ = "FIVE"
23: **end if**

---

**Algorithm 10** : Alphabet prediction

---

1: **// Purpose:** To predict the alphabets
2: **// Input:** $count\_defects$ and $area$
3: **// Output:** Predicted alphabet
4: - - - - - - - - - - - - - - - - - - - - - - - - - - - - - - - - - - - - - - - - - - - - - - - - - - - - - - - -
5: $area = \frac{area}{10000}$
6: **if** $count\_defects = 3 \ \wedge \ 1.5 \leq area \leq 2.0$ **then**
7:    $pred$ = "B"
8: **else if** $count\_defects = 1 \ \wedge \ 0.5 \leq area \leq 0.8$ **then**
9:    $pred$ = "C"
10: **else if** $count\_defects = 0 \ \wedge \ 0.5 \leq area \leq 0.8$ **then**
11:    $pred$ = "L"
12: **else if** $count\_defects = 0 \ \wedge \ 0.9 \leq area \leq 1.9$ **then**
13:    $pred$ = "P"
14: **else if** $count\_defects = 4 \ \wedge \ 0.9 \leq area \leq 1.9$ **then**
15:    $pred$ = "R"
16: **else if** $count\_defects = 0 \ \wedge \ 0.8 \leq area \leq 1.0$ **then**
17:    $pred$ = "W"
18: **else if** $count\_defects = 1 \ \wedge \ 0.9 \leq area \leq 1.9$ **then**
19:    $pred$ = "T"
20: **end if**

---

**Algorithm 11** : Word prediction

---

1: **// Purpose:** To predict the words
2: **// Input:** $count\_defects$, $area$
3: **// Output:** Predicted word
4: - - - - - - - - - - - - - - - - - - - - - - - - - - - - - - - - - - - - - - - - - - - - - - - - - - - - - - - -
5: $area = \frac{area}{10000}$
6: **if** $area \geq 3.0 \ \wedge \ count\_defects \geq 4$ **then**
7:    $pred$ = "DOCTOR"
8: **else if** $0.5 \leq area \leq 1.9$ **then**
9:    $pred$ = "THIS"
10: **else if** $1.9 \leq area \leq 2.5$ **then**
11:    $pred$ = "YOUR"
12: **else if** $1.9 \leq area \leq 3.5$ **then**
13:    $pred$ = "MINE"
14: **end if**

---

## A.3 TABLES

Table A1: Existing Sign Languages

| Sign Language | Region | Influenced by | Number of signers | Significance |
|---|---|---|---|---|
| American Sign Language (ASL)NIL (2017) | America, Canada, Southeast Asia and West Africa | French Sign Language, Martha's Vineyard Sign Language and other local sign languages | 250,000-500,000 | It is one of the easiest languages to learn because most of the signs were developed to mimic the actual word or phrase it is representing. |
| British Sign Language (BSL)Richard (2018) | UK | Evolved at Thomas Braidwood's schools for the deaf in the late 1700s and early 1800s | 150,000 | It spread to Australia and New Zealand, thus resulting in similarity of New Zealand Sign Language and Auslan (Australian Sign Language) |
| French Sign Language (LSF)Richard (2018) | France and Switzerland | Developed from the Parisian deaf community, and taught by Charles Michel de | 100,000 | It is one of the earliest European sign languages to gain acceptance by educators, and it influenced other sign languages like ASL, ISL, Russian Sign Language (RSL) and many more. |
| Brazilian Sign Language (Libras)Richard (2018) | Brazil | French Sign Language | 3 million | It was given official status by the Brazilian government in 2002. |
| Indo-Pakistani Sign Language (IPSL)Richard (2018) | South Asia | Hindi/Urdu, English, and BSL | Between 1.8 million and 7 million | Unlike ASL and many of the signed languages of Europe, IPSL does not have Classifier handshapes |

Table A2: Existing assistive technology for the speech impaired and the deaf and mute

| Features | Hand Talk Assistive technology | Uni Tablet | MotionSavvy | GnoSys |
|---|---|---|---|---|
| Category | AAC | AAC | AAC | AAC |
| Physical Appearance | Gloves | Tablet | Tablet App | Phone App |
| Attributes | Sign to Speech | Sign to Speech, Speech to Text | Sign to Speech, Speech to Text | Sign to Speech |
| Hardware Parts | Flex sensors, accelerometers and gyroscope | Tablet, case, Leap motion sensor | - | - |
| Software Parts | Arduino technology | Leap motion technology | Leap's 3D motion recognition technology | Neural networks and computer vision, cloud computing |
| Cost | Upto Rs. 10,000 | Upto 800 USD | 20 USD per month on subscription basis | 1 USD per day, 4 USD per week and 11 USD per month |
| Size | Size of the hand | 8" screen size | Fits the tablet screen | Fits the phone screen |
| Limitations | Size of gloves would vary, works only when the gloves are worn | Bigger size, costly | More storage space required, only for tablets | Only one way communication is possible, internet connection is required |

