# OpenReview forum: "URVoice: An Akl-Toussaint/ Graham- Sklansky Approach towards Convex Hull Computation for Sign Language Interpretation"
_ICLR.cc/2023/Conference — Submitted to ICLR 2023_

### Official Review · Reviewer_jVQ2 · 2022-10-22

**Confidence:** 5
**Correctness:** 1
**Technical Novelty And Significance:** 1
**Empirical Novelty And Significance:** Not applicable
**Recommendation:** 1

**Clarity, Quality, Novelty And Reproducibility:**

The method description is clear. There is no novelty in the proposed method. I think it is very easy to reimplement the method.

**Strength And Weaknesses:**

Strength:
- Sign language recognition is an important task.
- The author provided a detailed implementation of their method. I believe the proposed method could be easily re-implemented.

Weakness:
- There is no novelty in the proposed method. The proposed method uses very basic image processing techniques and very simple contour detection. The method is too naive.

- The author only evaluates their method on a small self-collected dataset. The evaluation dataset is collected from five subjects and only a few class images. There is no evidence showing the generally of the proposed method.

- The paper writing could be significantly improved. There are too many parts of the paper that need to be rewritten. I suggest the author find someone who got accepted a paper in ICLR before to give them professional suggestions in writing.

**Summary Of The Paper:**

This paper proposes a method for sign language hand gesture recognition. There is no novelty in the proposed method, and the author only evaluated their method on self-collected data.

**Summary Of The Review:**

I think this is a clear reject of the ICLR submission. I did not see the novelty of the proposed method and the evaluation was only done on a small self-collected dataset.

---

### Official Review · Reviewer_SEnB · 2022-10-23

**Confidence:** 5
**Clarity, Quality, Novelty And Reproducibility:** The novelty of this paper is insuffic…
**Correctness:** 2
**Technical Novelty And Significance:** 2
**Empirical Novelty And Significance:** 2
**Recommendation:** 3

**Strength And Weaknesses:**

*Strength:
The key strengths may lie in the proposed framework URVoice. As the author points out, the URVoice can take the visual / audio as input from the collocutor/ computer and converts it into gesture/ text as output for the signer.
* Weaknesses:
Frankly speaking, the title of this paper is very confusing. Why is it called URVoice? “convex hull computation” is simply too broad and vague. Although not explicitly being stated, this work focused on gesture recognition. With that being said, the main technical contributions seem to be the proposed framework to achieve translation of gesture to text/voice. However, with insufficient evidence to demonstrate the effectiveness of the proposed approaches than the state-of-the-art methods, the real impact that this work can bring to the community remains unclear to me.
After reading the whole paper, I strongly feel that the proposed algorithms seem to be trivial and found nothing that excited me. For gesture recognition, there have been many excellent literatures [1,2,3] can be reference.  For the transformation of visual/text signals to speech/visual signals, many multimodal literatures [4,5,6,7] have achieved this goal.

[1] H. Wang, P. Wang, Z. Song, and W. Li, “Large-scale multimodal gesture recognition using heterogeneous networks,” in Proceedings of the IEEE International Conference on Computer Vision (ICCV), Oct 2017.
[2] J. Wan, G. Guo, and S. Z. Li, “Explore efficient local features from rgb-d data for one-shot learning gesture recognition,” IEEE transactions on pattern analysis and machine intelligence, vol. 38, no. 8,
pp. 1626–1639, 2015
[3] G. Zhu, L. Zhang, L. Yang, L. Mei, S. A. A. Shah, M. Bennamoun,and P. Shen, “Redundancy and attention in convolutional lstm for gesture recognition,” IEEE transactions on neural networks and learning systems, vol. 31, no. 4, pp. 1323–1335, 2019.
[4] Jeonghun Baek, Geewook Kim, Junyeop Lee, Sungrae Park, Dongyoon Han, Sangdoo Yun, Seong Joon Oh, and Hwalsuk Lee. What is wrong with scene text recognition model comparisons? dataset and model analysis. In Proc. ICCV, 2019.
[5] Zhanzhan Cheng, Yangliu Xu, Fan Bai, Yi Niu, Shiliang Pu, and Shuigeng Zhou. Aon: Towards arbitrarily-oriented text recognition. In Proc. CVPR, 2018.
[6] Chen-Yu Lee and Simon Osindero. Recursive recurrent nets with attention modeling for OCR in the wild. In Proc. CVPR, 2016.
[7] Baoguang Shi, Mingkun Yang, Xinggang Wang, Pengyuan Lyu, Cong Yao, and Xiang Bai. Aster: An attentional scene text recognizer with flexible rectification. PAMI, 2018.

**Summary Of The Paper:**

The paper presents a vocalizer that achieves real-time translation of gesture to text/voice using convex hull as the computational geometry.

**Summary Of The Review:**

In conclusion, I think the novelty of this paper is insufficient, and the current version still has a lot of room for improvement, including the improvement of the algorithm and sufficient experimental support for the proposed technology.

---

### Official Review · Reviewer_eit5 · 2022-10-23

**Confidence:** 5
**Correctness:** 3
**Technical Novelty And Significance:** 1
**Empirical Novelty And Significance:** 1
**Recommendation:** 1

**Clarity, Quality, Novelty And Reproducibility:**

The paper is clear. There is no novelty in the methodology. The authors present several algorithms in the text, which might allow the reproducibility.

**Strength And Weaknesses:**

Main Strengths:
-The paper is fairly well-written and organized.

Main Weaknesses:
Unfortunately, I believe the paper is out of the scope of the conference. It is focused on the architecture of a device (the URVoice), which is based on fundamental geometrical and computer vision algorithms. The authors review such fundamental methodology, but they do not present any related state-of-the-art literature on sign language recognition. Finally, the paper does not propose any novel contribution. The experiments are also performed on limited datasets.

**Summary Of The Paper:**


The paper introduces the URVoice, an Augmentative and Alternative Communication (AAC) device which is supposedly a technological innovation. It is based on the Akl-Toussaint/Graham-Sklansky convex hull algorithms for sign language recognition.

**Summary Of The Review:**

Unfortunately, I believe the paper is out of the scope of the conference. I suggest the authors submit their work to another conference or journal focused on technological innovation on such a kind of devices.

---

### Official Review · Reviewer_aLeh · 2022-10-24

**Confidence:** 4
**Correctness:** 1
**Technical Novelty And Significance:** 3
**Empirical Novelty And Significance:** 3
**Recommendation:** 1

**Clarity, Quality, Novelty And Reproducibility:**

The paper could acknowledge and appreciate prior work more. For example, citing related work is encouraged. Its reference list is very short and mostly outdated. For example, advances in Indian natural language processing and sign language processing in 2020s have not been recognized here; many original studies and literature reviews published in the last 5 years are available but none of them have been cited in this paper.

The paper does not seem to be using recommended English terminology either. My understanding is that the peak advocacy and information organizations recommend using the terms of the Deaf and Hard-of-Hearing communities.

The broader impact and ethical considerations of this study should be discussed in further depth. I find the current addressing insufficient and more generally, the paper does not seem to follow the discipline traditions in the expected structure, order, and depth/length in addressing the why and so what of the study.

**Details Of Ethics Concerns:**

The paper unfortunately suffers from severe issues in addressing human research ethics. Although it created and used two different datasets including data from human subjects, it did not describe human research ethics at all. I would have expected that obtaining ethical approvals, research permissions, and participants' informed consent to be described in general and in particular because this study focused on a particularly vulnerable cohort of people using sign language.

The paper does not seem to be using recommended English terminology either. My understanding is that the peak advocacy and information organizations recommend using the terms of the Deaf and Hard-of-Hearing communities.

**Strength And Weaknesses:**

The paper proposes a new communication device to "translate" from Indian sign language gestures to text and/or voice. Both Indian natural language processing and sign language processing are severely under-resourced and societally significant, hence making this contribution even more substantial.

The paper unfortunately suffers from severe issues in addressing human research ethics. Although it created and used two different datasets including data from human subjects, it did not describe human research ethics at all. I would have expected that obtaining ethical approvals, research permissions, and participants' informed consent to be described in general and in particular because this study focused on a particularly vulnerable cohort of people using sign language.

**Summary Of The Paper:**

The paper proposes a new communication device to "translate" from Indian sign language gestures to text and/or voice. Both Indian natural language processing and sign language processing are severely under-resourced and societally significant, hence making this contribution even more substantial.

**Summary Of The Review:**

Without addressing research ethics and other concerns related to relevant human subjects more broadly (e.g., the use of recommended terminology and discussion of the broad impact of this study), I can only recommend a strong rejection of this study.

---

> ### Comment · Reviewer_aLeh · 2022-11-21
> **No author response to acknowledge**
>
> No author response to acknowledge: Authors have not responded to my review above.

---

### Decision · Program_Chairs · 2023-01-20

**Decision:**

Reject

**Justification For Why Not Higher Score:**

The paper's main contribution could be the description of a vocalization system for an under-resourced sign language, which makes it a good fit for a conference on under-resource language technologies, but less good a fit for ICLR. The paper also ignores requirements such as ethics discussion, and authors did not submit a rebuttal to the reviewers' questions.


**Justification For Why Not Lower Score:**

n/a

**Metareview: Summary, Strengths And Weaknesses:**

Summary

The paper proposes a new real-time communication device to "translate" (vocalize) from Indian sign language gestures to text and/or voice. The work is based on the known Akl-Toussaint heuristic and Graham-Sklansky convex hull algorithm, compared to other known approaches, and thus presents no technological innovation. Evaluation happens on self-collected data.

Strengths

Both Indian natural language processing and sign language processing are severely under-resourced and societally significant, making this contribution potentially substantial. The paper is by and large well-written.

Weaknesses

The paper presents no technical novelty, and thus does not appear to be a good match to the conference. The authors do not comment on potential ethical/ societal aspects of their work, which is a requirement. It is unclear how the technical content of the paper compares to other well-known techniques, i.e. if it has merit in terms of SOTA performance. The paper cites a limited amount of other relevant work.


**Summary Of Ac-Reviewer Meeting:**

n/a